# Phagocytosis in Marine Coccolithophore *Gephyrocapsa huxleyi*: Comparison between Calcified and Non-Calcified Strains

**DOI:** 10.3390/biology13050310

**Published:** 2024-04-30

**Authors:** Jiayang Ye, Ying Wang, Qian Li, Sarfraz Hussain, Songze Chen, Xunying Zhou, Shengwei Hou, Yuanyuan Feng

**Affiliations:** 1School of Oceanography, Shanghai Jiao Tong University, Shanghai 200030, China; yejiayang@sjtu.edu.cn (J.Y.); liqian00@sjtu.edu.cn (Q.L.); 2Shanghai Key Laboratory of Polar Life and Environment Sciences, Shanghai Jiao Tong University, Shanghai 200030, China; housw@sustech.edu.cn; 3Key Laboratory of Polar Ecosystem and Climate Change, Shanghai Jiao Tong University, Ministry of Education, Shanghai 200030, China; 4Department of Ocean Science & Engineering, Southern University of Science and Technology, Shenzhen 518055, China; 5Shenzhen Ecological and Environmental Monitoring Center of Guangdong Province, Shenzhen 518049, China

**Keywords:** coccolithophore, *Gephyrocapsa huxleyi*, life stage, haploid, diploid, calcification, phagocytosis

## Abstract

**Simple Summary:**

The nutrient strategy is a significant part of *Gephyrocapsa huxleyi*’s physiology. Unlike other studies mainly focusing on photosynthesis, we conducted a short-term incubation to test the difference in phagocytosis capacity between diploid calcified *G. huxleyi* RCC1266 and haploid non-calcified *G. huxleyi* PML B92/11 in different light regimes. The results suggested that a higher percentage has been found in the non-calcified strain as compared to the calcified strain, no matter whether it was under light or in the dark. Additionally, the non-calcified strain showed a greater percentage in the dark than under light.

**Abstract:**

Coccolithophores play a significant role in marine calcium carbonate production and carbon cycles, attributing to their unique feature of producing calcareous plates, coccoliths. Coccolithophores also possess a haplo-diplontic life cycle, presenting distinct morphology types and calcification states. However, differences in nutrient acquisition strategies and mixotrophic behaviors of the two life phases remain unclear. In this study, we conducted a series of phagocytosis experiments of calcified diploid and non-calcified haploid strains of coccolithophore *Gephyrocapsa huxleyi* under light and dark conditions. The phagocytosis capability of each strain was examined based on characteristic fluorescent signals from ingested beads using flow cytometry and fluorescence microscopy. The results show a significantly higher phagocytosis percentage on fluorescent beads in the bacterial prey surrogates of the non-calcified haploid *Gephyrocapsa huxleyi* strain, than the calcified diploid strain with or without light. In addition, the non-calcified diploid cells seemingly to presented a much higher phagocytosis percentage in darkness than under light. The differential phagocytosis capacities between the calcified diploid and non-calcified haploid *Gephyrocapsa huxleyi* strains indicate potential distinct nutritional strategies at different coccolithophore life and calcifying stages, which may further shed light on the potential strategies that coccolithophore possesses in unfavorable environments such as twilight zones and the expanding coccolithophore niches in the natural marine environment under the climate change scenario.

## 1. Introduction

Coccolithophores are cosmopolitan unicellular algae with mineralized calcium carbonate scales known as “coccoliths” [1] with a global scale of distribution from tropical to subpolar oceanic regions [2,3]. Due to their involvement in both photosynthesis and calcification, this group is considered a crucial phytoplankton functional group in the marine carbon cycle [4,5]. It is estimated that coccolithophores contribute 7~10% [6], and occasionally as much as 40% of total carbon fixation [7], in open ocean ecosystems. Additionally, they are one of the primary contributors to phytoplankton blooms and form the “white water”, resulting from the scattering of coccoliths [8,9]. The calcareous shells of coccolithophores increase the density of the cells and their sinking rates, making coccolithophores sink faster in the water column. By changing the buoyancy of coccolithophores, coccoliths act as a ballast in this process, which is referred to as the “ballast effect” [10,11]. The ballast effect plays a crucial role in transporting organic carbon captured in the surface ocean to the deeper ocean, thus modulating the efficiency of marine biological carbon pump [12]. Coccolithophore blooms greatly increase both POC (particle organic carbon) and PIC (particle inorganic carbon) export to the deep ocean [13], and the deposited coccoliths eventually contribute to the formation of calcareous ooze in deep-sea sediments. The most abundant and widely distributed coccolithophore species in the ocean is *Gephyrocapsa huxleyi*, formerly known as *Emiliania huxleyi* [14,15,16], which has been widely used as a model organism to study biogenic calcification and in the marine global carbon cycle [16,17].

The biogenic calcification process is one of the most distinguishing characteristics of coccolithophores, which employ bicarbonate and calcium ions from seawater to produce intricate calcium carbonate and release carbon dioxide and water molecules as byproducts. The coccoliths are produced within a Golgi-derived calcified compartment, called the coccolith vesicle [18]. Based on the presence of coccoliths, coccolithophores are divided into two distinct morphotypes, calcified and non-calcified. However, morphotype transition was also observed in certain types of coccolithophores species during long-term culture in laboratory settings [19] or after viral infections [20]. For *G. huxleyi* species, it has been reported that special treatments in solid media incubation could trigger the phase transition between diploid calcified and haploid non-calcified life stages [21]. A recent study on the joint proteomic and genomic investigation provided more information on the coccolith-related proteins [22]. Despite these, the ecological roles of biogenic calcification have not been fully resolved. Some studies revealed that calcification may provide the cells grazing protection, viral/bacterial attack protection [23], light environment modification [24], and trash carbonate consumption [19]. Other contradictory studies showed that calcified *G. huxleyi* was more easily grazed [25], and the transition to a non-calcified form was observed when calcified cells were exposed to the viruses [20].

Marine coccolithophores in general utilize a “haplo-diplontic” life cycle strategy, in which the cells are able to divide in both haploid (with one set of chromosomes) and diploid phases (with two sets of chromosomes) [26]. These different life cycle phases of coccolithophores may show various forms of coccolith size and morphology. For coccolithophore *G. huxleyi*, the cells present three life stages, C-cell for the non-motile diploid form with highly calcified coccoliths, N-cell for the non-motile diploid form without coccoliths, and S-cell for the motile haploid form only with organic body scales but no coccoliths [19,27]. The diploid C-cell and N-cell show no flagella, while the haploid S-cell has two flagella and associated flagellar bases [28,29,30].

Physiological differences between calcified and non-calcified *G. huxleyi* strains in different life stages have been reported, including the production of transparent exopolymer particles (TEPs), growth rate, and cellular element content [31,32]. The adaptation capacity to irradiance has also been reported to be different between the two morphotypes. For example, the non-calcified strain exhibited photoinhibition under a moderate level of irradiance of above 400~500 μmol m^−2^ s^−1^ [24]. The photosynthesis of the non-calcified haploid strain RCC 1217 exhibited greater sensitivity to high visible light levels compared to the calcified strain PML B92/11, but demonstrated resistance to ultraviolet radiation (UVR)-induced inhibition, demonstrating a capacity to adapt to a specific ecological niche [33]. Previous studies have indicated that cellular organic contents between the two morphotypes may differ, but the results have been inconsistent across experiments. The cellular particulate organic nitrogen (PON) and particulate organic carbon (POC) contents increased to a much higher level in the calcified *G. huxleyi* strain than in the non-calcified strain by increased CO_2_ levels [34]. However, in another study, cellular POC production and cell size decreased significantly in a non-calcified *G. huxleyi* strain under elevated *p*CO_2_, while they remained unchanged in the calcified strain [35]. In addition, calcified and non-calcified *G. huxleyi* strains may have different defense capabilities against predators. In terms of short-term ingestion rate, the non-calcified *G. huxleyi* showed a stronger defense response, while the calcified strain showed no defense response [36]. Even during blooms, there is significant fluctuation in bloom timing and cell abundance of these two morphotypes [37].

In terms of nutritional strategies, coccolithophores tend to have the potential of mixotrophy, such as phagotrophy and osmotrophy [38,39,40]. There is increasing evidence that many phytoplankton organisms are mixotrophs, i.e., they possess both the ability of photoautotrophy and, to varying degrees, to ingest prey by phagotrophy. Mixotrophic organisms, which combine photoautotrophy and phagotrophy in a single cell, are widespread in marine ecosystems. This dual trophic strategy appears to be particularly advantageous when light or inorganic nutrients are limited. In a recent study, phagotrophy was investigated in four species of coccolithophores, including the non-calcified *G. huxleyi* (RCC 1216/RCC1217), and it was found that phagocytic activity was lower in *G. huxleyi* compared to the other species and no clear difference in ingestion rate was detected between the two phases of *G. huxleyi* [38]. There were also different results in another study in the transcriptome analysis and the microscopy observations. The transcriptome analysis represented that higher abundances of transcripts related to endocytic and digestive machinery can be found in the calcified strain compared to the non-calcified strain, whereas the microscopy observations revealed that phagocytotic particle uptake could be found in the late stationary phase of both the calcified and non-calcified strains [41].

In addition, calcified and non-calcified *G. huxleyi* strains showed significant differences in omics. Transcriptomic analyses on calcified *G. huxleyi* CCMP371 cultivated in different calcium concentrations indicate that the differentially expressed genes (DEGs) were mainly enriched in the areas of membrane component, secondary metabolic pathway, signal transduction, ABC transporters, and protein synthesis [42,43]. Transcriptomic investigations of the calcified strain RCC1216 and non-calcified strain RCC1217 of *G. huxleyi*, demonstrated distinct differences in the regulation of genome expression, proteome maintenance, and metabolic processing between two life-cycle stages [41]. However, another study using the same strains of *G. huxleyi* showed that signal transduction and motility genes are among the most significantly up-regulated genes in the non-calcified strain compared to the calcified strain, and the significantly calcified specific genes are related to Ca^2+^, H^+^, and HCO_3_^−^ pumps, which are consistent with their features [27].

Due to these divergent physiological differences in different coccolithophore life stages/morphology, the coccolithophore ecology can be affected by coccolith morphology and the ploidy of the cells [44]. As a consequence, the calcified and non-calcified coccolithophores could occupy distinct ecological niches and play different roles in the marine carbon cycle [26]. Here, to compare the phagocytosis capacity between these two strains, we carried out fluorescence beads digestion experiments on calcified *G. huxleyi* RCC1266 and non-calcified *G. huxleyi* PML B92/11. Following the previous studies and the distinct features between the two strains, we hypothesized that the haploid non-calcified strain has a greater capacity to phagocytose compared to the diploid calcified strain.

## 2. Methods

### 2.1. Stock Culture Conditions

*G. huxleyi* RCC1266 (calcified strain, isolated from continental shelf waters around Ireland) and PML B92/11 (originated from calcified strain and transformed to non-calcified strain in a lab incubation lasting years, isolated from the North Sea) were used in the research. Both were cultured in natural seawater enriched with NaNO_3_, NaH_2_PO_4_·2H_2_O, trace metals, and vitamins at the concentration of f/20 medium (ten times the dilution of f/2) [45]. The stock cultures were kept in an incubator (GXZ-280D, Ningbo Jiangnan, Ningbo, China) at a constant temperature of 15 °C and irradiance level of 100 μmol m^−2^ s^−1^ (light: dark = 12 h:12 h) in the School of Oceanology, Shanghai Jiao Tong University.

### 2.2. Morphology, Cell Size, and Ploidy Examination

The morphology of the strains and cell sizes were examined under an optical microscope (CH20BIMF200, Olympus, Tokyo, Japan) at a magnification of ×400. A flow cytometer (CytoFLEX, Beckman Coulter, Brea, CA, USA) was used to detect differences in cell diameter and ploidy. For flow cytometry pretreatment for ploidy distinctions, photoperiodic cells were fixed with 0.5% glutaraldehyde (Macklin, Shanghai, China). Cellular DNA was stained using 4′,6-diamidino-2-phenylindole (DAPI, Sigma-Aldrich, Darmstadt, Germany). Cells were identified via red fluorescence (Chl a, γem ~690 nm). The cell size and internal complexity were reflected by forward scatter (FSC) and side scatter (SSC), respectively. The cellular DNA content was indicated by blue fluorescence (stained nucleic acid, γem ~450 nm).

### 2.3. Phagocytosis Experiment

In terms of strain growth conditions for phagocytosis, *G. huxleyi* RCC1266 and *G. huxleyi* PML B92/11 cultures were grown to the exponential growth period. The phagocytic capability was detected by fluorescence microscopy and flow cytometry, respectively.

For fluorescence microscopic observation, fluorescent beads of 1 μm (L4655, Sigma-Aldrich, Darmstadt, Germany) and 0.5 μm (L5530, Sigma-Aldrich, Darmstadt, Germany) were both added to the culture. The final concentrations of both sized beads were 2.5 × 10^5^ mL^−1^. Then, the cultures were placed in an incubator at 15 °C for 2 h under light (100 μmol m^−2^ s^−1^). At the end of the incubation, the cells were fixed with a final concentration of 0.5% glutaraldehyde (Macklin, Shanghai, China) for 10 min. After the incubation, the cells were stained with the ultimate concentration of DAPI (Sigma-Aldrich, Darmstadt, Germany) at 0.02 μg mL^−1^ and filtered onto 2 μm polycarbonate membranes (Millipore, Darmstadt, Germany). The filter membrane was then fixed on a slide under the cover glass and observed under an inverted fluorescence microscope (BX61, Olympus, Tokyo, Japan).

For the flow cytometric examination, only 1 μm fluorescent beads were added at the final concentrations of 2.5 × 10^5^ mL^−1^ due to the weak fluorescence of 0.5 μm beads. Then, the cultures were placed in an incubator at 15 °C for 2 h under both light (100 μmol m^−2^ s^−1^) and dark conditions. At the end of the incubation, the cells were fixed in the same way as fluorescence microscopic observation. After that, flow cytometry was used to detect the digestion percentage difference between the two strains. The samples were stained using DAPI (Sigma-Aldrich, Darmstadt, Germany) with a final concentration of 4 μg mL^−1^. Pure cultures of two strains without fluorescent beads’ addition, which undergo the same treatments as the samples, and 1 μm pure fluorescent beads were used as blank controls based on red fluorescence (γem ~690 nm), blue fluorescence (γem ~450 nm), and green fluorescence (γem ~525 nm). Furthermore, samples were treated using Poloxamer 188 solution (Sigma-Aldrich, Darmstadt, Germany) to reduce adhesion between cells and beads. To avoid beads being stuck by coccoliths, 10% HCl solution addition for 1 min was used to dissolve the coccoliths, then 1 M NaHCO_3_ solution was used to neutralize the samples. A flow cytometer (CytoFLEX, Beckman Coulter, Brea, CA, USA) was used for fluorescence signal acquisition and the phagocytosis percentage calculation. The cells with and without beads were identified by red fluorescence (γem ~690 nm) and green fluorescence (γem ~525 nm) [46].

The phagocytosis percentages (*PP*) were calculated using the following equation:$$PP=\frac{N_{with}}{N_{with}+N_{without}}\times 100\%$$
where *N_with_* and *N_without_* are the cells with and without the fluorescent beads, respectively.

### 2.4. Statistical Analyses

To test the cell diameter difference between diploid calcified and haploid non-calcified *G. huxleyi*, we used a parametric *t*-test for pairwise comparison on a total of 20 cells in each strain after the Shapiro–Wilk test to prove its normal distribution. For comparison of the phagocytosis capacities between the two *G. huxleyi* strains under different light regimes, significance was determined through a two-way ANOVA (Analysis of Variance) test at *p* = 0.05 level. For different strains’ results, we tested the hypothesis that the ploidy had an effect on the phagocytosis by comparing the phagocytosis percentages during the 2 h incubation time of the diploid calcified strain with the haploid non-calcified strain. For the same strain in different light regimes, we tested the hypothesis that darkness had a great effect on the phagocytosis percentages by comparing the percentages during the 2 h incubation time of the same strain in darkness with that in light. All statistical analyses and graphical representations of data were performed using GraphPad Prism 9.5.0 (Boston, MA, USA).

## 3. Results

### 3.1. Differences in Morphotype and Ploidy

The microscopic image of *G. huxleyi* RCC1266 showed that the cells were calcified with extracellular coccoliths, with significantly larger cell size compared to *G. huxleyi* PML B92/11 (Figure 1a–c). The average cell diameters of *G. huxleyi* RCC1266 and *G. huxleyi* PML B92/11 were 5.66 ± 0.50 μm and 4.57 ± 0.54 μm, respectively.

Flow cytometry analysis unveiled differences in both cell size and ploidy between the calcified and non-calcified strains. The calcified strains demonstrated a higher signal of side scatter (SSC) in comparison to their non-calcified counterparts (Figure 2a). Furthermore, a notable difference in cell ploidy was observed between the calcified and non-calcified strains, as evidenced by their DNA content. Examining fluorescence characteristics, the mean fluorescence intensity in blue fluorescence (γem ~450 nm) was much higher in the calcified strain (125,952) than in the non-calcified strain (65,225), with the calcified strain exhibiting DNA content approximately twice that of the non-calcified strain (Figure 2b).

### 3.2. Phagocytosis Capacity

#### 3.2.1. Fluorescence Microscopic Examination

The fluorescence microscopic examination performed after the phagocytosis incubation experiment suggested that both the calcified *G. huxleyi* and non-calcified *G. huxleyi* strains were able to take up beads of both sizes within 2 h of incubation time under light condition (Figure 3).

#### 3.2.2. Flow Cytometric Examination

The flow cytometry examination showed significantly different phagocytosis percentages between *G. huxleyi* RCC1266 and *G. huxleyi* PML B92/11 under both light and dark conditions (Figure 4). The non-calcified strain *G. huxleyi* PML B92/11 exhibited much higher phagocytosis rates than those of the calcified strain, with about 4.6- and 3.3-times values of those in the calcified strain in light and dark conditions, respectively. In addition, for the non-calcified strain PML B92/11, a significantly higher phagocytosis percentage was observed under dark condition (2.51% ± 0.31%) than under light (1.71% ± 0.26%). On the contrary, there was no significant difference in the phagocytosis percentage between light and dark conditions for the calcified strain (Figure 4, Appendix A).

## 4. Discussion

### 4.1. Long-Term Phagocytosis Experiments Still Need to Be Completed

Different from long-term environment simulations, our experiments focused on the short-term consequences of the phagocytosis between the calcified diploid and non-calcified haploid *Gephyrocapsa huxleyi* by comparing phagocytosis percentages, which only indicate the potential nutritional strategy change in a limited time. In addition, in the experiment, we did not measure the percentage and amount of energy acquired from both photosynthesis and phagocytosis, which makes it hard to quantify if it is enough for the survival of *G. huxleyi* in the long term. In our results, light works as a barrier in the phagocytosis of non-calcified *Gephyrocapsa huxleyi*, which showed a similar conclusion to McKie-Krisberg’s study [47]. However, Sander’s research on the mixotrophic chrysophyte *Poterioochromonas malhamensis* showed that the light intensity did not present an effect on the phagotrophy [48], while Pang and Hansen’s studies found that increasing light intensity could enhance the ingestion rate [49,50]. As there are diverse views on the effect of light on the ingestion rate, we assume that the effect might vary from the species and more studies should be completed.

### 4.2. Physiological Differences between Calcified and Non-Calcified G. huxleyi Strains and the Potential Ecological Niches

Marine coccolithophores can utilize a distinctive haplo-diplontic life cycle and the cells are able to divide in both life cycle phases, which potentially expands the spectrum of ecological niches in the marine environment [51,52]. In the present study, the ploidies of the two *G. huxleyi* strains were detected using flow cytometry, based on the hypothesis that within one species the DNA content per cell is relatively constant and depends on the state of ploidy and the basic amount of DNA per set of ploidies [53,54,55]. The results revealed that the calcified strain *G. huxleyi* RCC1266 was in the diploid phase, while the non-calcified strain *G. huxleyi* PML B92/11 was in the haploid phase (Figure 2b).

Between the diploid calcified and haploid non-calcified strains, there are significant physiological differences. The non-calcified *G. huxleyi* PML B92/11 used in the present study exhibited a higher growth rate compared to the calcified *G. huxleyi* RCC1266 at 15 °C under an irradiance level of 100 μmol m^−2^ s^−1^ [32]. Due to the presence of coccoliths, the calcified *G. huxleyi* cells were found to possess much higher sinking rates compared to the naked non-calcified cells (Appendix A) [32]. In addition, in the study that used the same strains as ours, the non-calcified strain *G. huxleyi* PML B92/11 was found to have a smaller cell diameter, and lower cellular particulate organic carbon (POC) and cell protein contents than the calcified strain RCC 1266 (Appendix A) [32]. Furthermore, differences in induced defensive capabilities, with only non-calcified strains displaying a defense response to the predator-exposed experiments under nutrient-replete conditions, suggested their different contributions to the marine food web [36]. Calcified *G. huxleyi* strains manifest their unique ecological advantages due to the stabilized microenvironment provided by the coccoliths when exposed to a combination of environmental stressors [56], perhaps explaining their prevalence in the natural environment.

Although numerous advances have been made in the ecological functions and responses to climate changes of marine coccolithophores, particularly common for calcified strains of the cosmopolitan species *Gephyrocapsa huxleyi* [57,58,59], studies on the ecology of non-calcified coccolithophore remain relatively limited. A research showed that the non-calcified strain could appear in small proportions at the end of coccolithophore blooms when *G. huxleyi* viruses’ (EhVs’) abundance increased [37]. Despite this, the haploid non-calcified coccolithophore strains are more difficult to spot under microscopes due to their indistinct morphology and smaller cell size, in contrast to the calcified strains. Therefore, there are few studies on directly isolated non-calcified *G. huxleyi* in the field [20].

### 4.3. Phagocytosis in Calcified and Non-Calcified Gephyrocapsa huxleyi

Phagocytosis refers to the process of ingesting and internalizing particulate matters larger than 0.5 μm in diameter by predatory or mixotrophic organisms, which is a complex biological process involving the coordinated regulation of various genes and proteins [60]. It can be divided into several stages: particle detection mediated by dedicated receptors in the cells, internalization process activation, phagosome formation, and maturation of the phagosome to transform it into a phagolysosome [61]. In this case, the genes in KEGG pathways like “phagosome” (pathway ID: ehx04145) are thought to be related to this complex biological process. Additionally, genes like AP2-related subunit and SNARE-type sorting factors have been reported to be endocytosis and phagocytosis-related in a previous study [41].

A previous study showed that phagocytotic particle uptake happened in the late stationary phase in both calcified and non-calcified strains of *G. huxleyi* [41]. Similarly, our study also found that both strains were capable of bead ingestion during a similar growth phase. Moreover, our results showed that the *G. huxleyi* non-calcified phase had a higher phagocytosis percentage than the calcified phase, which is probably related to cell morphology. For *G. huxleyi*, the haploid non-calcified phase has flagella and therefore displays strong mobility [62], which was believed to be relevant to the mixotrophic behavior [63,64].

Furthermore, our flow cytometry results indicate that darkness could significantly affect the phagocytosis capability of the haploid non-calcified phase (Figure 4). It is believed that coccolithophores are able to survive in low light or even dark conditions through osmotrophy [40,65] and phagotrophy [38]. Our research further supported the potential of phagocytosis as a nutritional source for *G. huxleyi*, especially in the dark.

Phagocytosis may impact the roles that organisms play in the food web, ecosystem, and carbon cycle [66,67]. The findings of the present study provide additional evidence for the role of *G. huxleyi* as a consumer in the marine food web. Previous studies have primarily focused on the contribution of mixotrophic organisms to carbon flow mostly at the community level, with significantly fewer studies on the species level [68]. Our results further reveal that even within a single species, the nutritional strategies can be different at different life stages or under different light regimes, thus having differential consequential impacts on the marine carbon cycle. This stresses the importance of future research to pay attention to smaller scales, even potentially down to the level of single cells using methods like nanoSIMS [69] and microfluidics with *in situ* Raman spectroscopy [70] on the ecologically important species in the marine ecosystem.

### 4.4. Survival under Light Limitation

Light plays an essential role in regulating the ecophysiology of *G. huxleyi*, such as coccolith formation and bloom occurrence [71]. However, light regimes may vary to a large extent in the marine environment where *G. huxleyi* dwells. For example, light is limited in the twilight zone when the cells sink 200 to 1000 m. Some extreme events in the geological periods, such as volcanic eruptions and meteorite impacts, could also lead to low light or even darkness due to dust input [72]. Furthermore, *G. huxleyi* has been found to expand poleward under the climate change scenario, which can lead to shorter light duration or even total light limitation in winter [73].

Some studies suggest that mixotrophy, both phagotrophy and osmotrophy, might act as a potential strategy for survival under low light or even darkness conditions, in historical periods, such as the end-Cretaceous period [74,75], with low nutrient concentrations. Our study observed that *G. huxleyi* possessed a higher phagocytosis capability in the haploid non-calcified phase than in the diploid calcified phase. On this account, *G. huxleyi* may transform into the haploid non-calcified phase of the life cycle to survive in dark or nutrient-limited conditions. This mechanism may provide *G. huxleyi* more flexibility to adapt to other environments, thus expanding its potential ecological niches.

### 4.5. Oceanic and Ecological Implications

By extrapolating these observed differences at the species level into the broader marine environment, our study may have further implications for understanding the different roles that diploid calcified and haploid non-calcified phases of coccolithophores play in the marine biogeochemical cycles and the potential ecological niches of these different phases (Figure 5). Based on our current knowledge, marine coccolithophores primarily exist in calcified form and play a crucial role in the organic carbon pump and carbonate counter pump in the ocean [19]. In the biogeochemical processes that marine coccolithophores participate in, coccoliths aid through the ballast effect, facilitating faster sinking, reducing the extent of remineralization in the mid-ocean layers, and contributing to the formation of coccolith ooze in the deep sea [11,76]. In contrast to the calcified strain, the non-calcified strain, lacking coccoliths and possessing two flagella [28,29,30], does not participate in the carbonate counter pump with much less contribution to the carbon pump into depth. However, in the deeper twilight zone, the phagotrophic nutritional pathway observed in the present study in addition to the previously reported osmotrophic pathway [40,65], may together benefit coccolithophore survival under energy-limited low light and dark conditions.

## 5. Conclusions

In summary, here we reported that the haploid non-calcified phase of *G. huxleyi* has a higher greater phagocytotic capacity than the diploid calcified phase. In addition, darkness favored the phagocytosis behavior of *G. huxleyi* of both phases rather than the light. These findings provide new insights into the potential nutritional strategies of marine coccolithophores in different light regimes and the environmental niches of different morphological phases in the marine environment. Further studies on the molecular mechanisms regulating the *G. huxleyi* mixotrophic behaviors and phase transition in *G. huxleyi* cells with the same genetic background are still necessary for a comprehensive understanding of the nutritional strategies and the roles in the marine carbon cycle of this model species.

## Figures and Tables

**Figure 1 biology-13-00310-f001:**
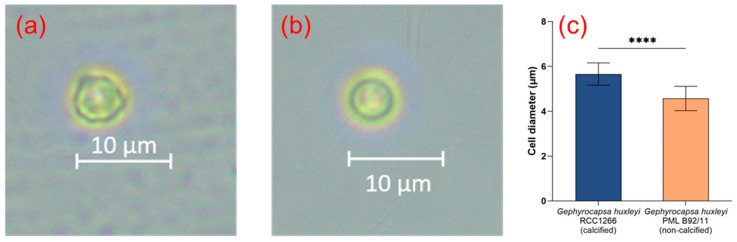
The optical microscope images showing the different morphological features of the *Gephyrocapsa huxleyi* calcified strain RCC1266 (**a**) and *Gephyrocapsa huxleyi* non-calcified strain PML B92/11 (**b**) in the stocking conditions. The average cell diameter difference between *Gephyrocapsa huxleyi* calcified strain RCC1266 and non-calcified strain PML B92/11 with a total of 20 cells of each strain measured by optical microscopy (**c**). ****: *p*-value < 0.0001.

**Figure 2 biology-13-00310-f002:**
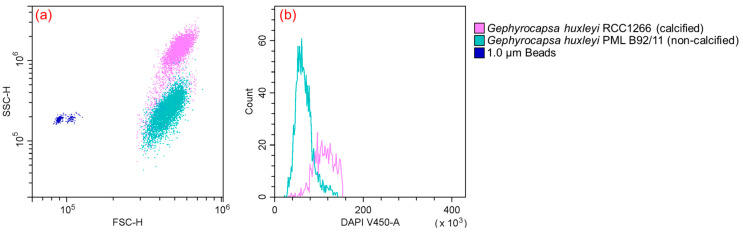
Flow cytometry plot (**a**) showing different features between *Gephyrocapsa huxleyi* calcified RCC1266 (pink) and *Gephyrocapsa huxleyi* non-calcified PML B92/11 (green) identified by γem ~690 nm compared to the 1 μm bead standards (blue) identified by γem ~525 nm based on forward scatter and side scatter. Flow cytometry plot (**b**) in γem ~450 nm showing different DNA contents between *Gephyrocapsa huxleyi* calcified RCC1266 (pink) and non-calcified PML B92/11 (green).

**Figure 3 biology-13-00310-f003:**
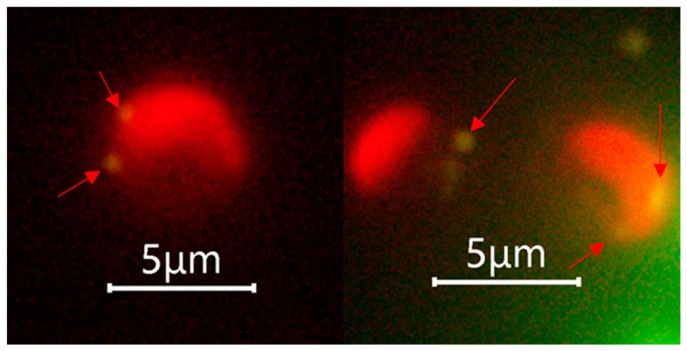
Selected fluorescent microscopic images showing phagocytosis evidence of *Gephyrocapsa huxleyi* under light. Arrows indicate the position of ingested prey surrogates, and only 0.5 μm beads were shown as they achieved the best aesthetic result with suitable fluorescence intensity.

**Figure 4 biology-13-00310-f004:**
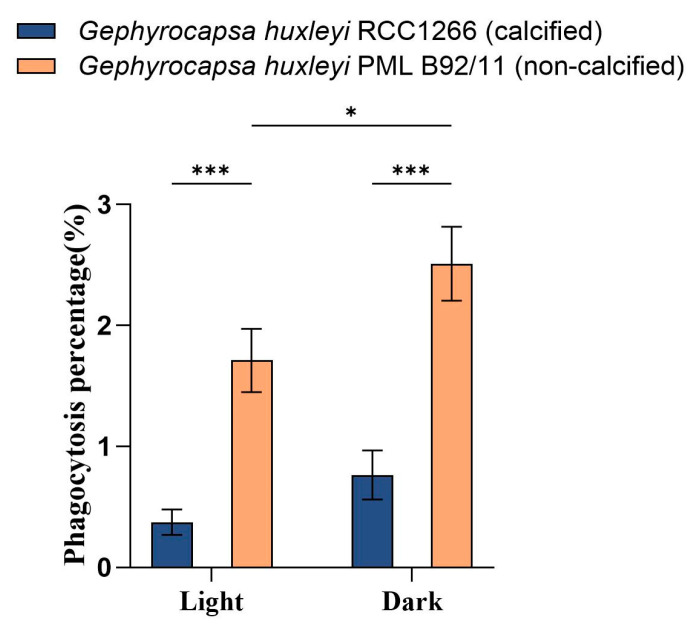
Phagocytosis percentages based on flow cytometric examination on the exponential growth phase of *Gephyrocapsa huxleyi* calcified strain RCC1266 and non-calcified strain PML B92/11 under light (100 μmol m^−2^ s^−1^) and dark in the 2 h incubation time at 15 °C. A two-way ANOVA test was used to determine the significance. *: 0.01 < *p*-value < 0.05, ***: *p*-value < 0.001. Error bars denote standard error, *n* = 3.

**Figure 5 biology-13-00310-f005:**
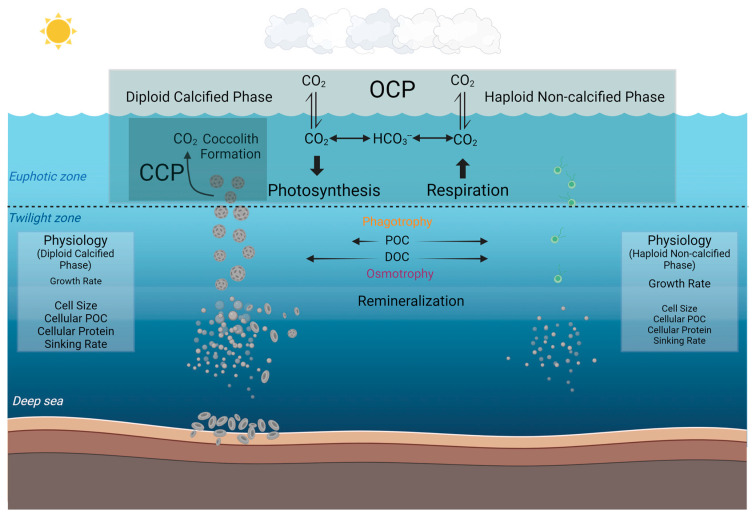
Conceptual model illustrating the different roles that calcified and non-calcified phases of *Gephyrocapsa huxleyi* may play in the marine carbon cycle by synthesizing the results of the present study and previous results. The organic carbon pump (OCP) denotes the organic carbon production through photosynthesis, and the carbonate counter pump (CCP) denotes the calcification process of calcium carbonate (coccoliths) production [77,78]. In our investigation, the particle organic carbon can be phagocytosed to a higher percentage in the non-calcified phase than the calcified phase in the dark as compared to the light condition. Dissolved organic carbon (DOC) has been reported to be utilized by coccolithophores with osmotrophy in low light [39,65]. The physiological differences in terms of growth rate, cell diameter, cellular POC, cellular protein, and sinking rate are also listed (The relative font size indicates the relative magnitude of each physiological metric between the two phases) [32]. This figure was created at BioRender.com (accessed on 2 April 2024).

## Data Availability

Data are contained within the article and Appendix A.

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
