# Peer review of "Phagocytosis in Marine Coccolithophore Gephyrocapsa huxleyi: Comparison between Calcified and Non-Calcified Strains"

_biology, 2024, doi:10.3390/biology13050310_

Round 1

Reviewer 1 Report

Comments and Suggestions for Authors

Coccolithophores, vital in marine ecosystems, produce calcium carbonate. Their life cycle and nutrient acquisition vary. This study tested phagocytosis in calcified diploid and non-calcified haploid strains of Emiliania huxleyi under light and dark conditions. Non-calcified haploid strains showed higher phagocytosis rates. This suggests different nutritional strategies across coccolithophore life stages, vital for adaptation in changing marine environments.

COMMENTS:

1. Emiliania huxleyi was named Gephyrocapsa huxleyi as mentioned in a recent publication (https://pubmed.ncbi.nlm.nih.gov/37983837/). Please check and rectify if needed to avoid confusion later.

2. How does the current study agree or disagree with a previous study using fluorescently labelled bacteria to assess phagocytosis in different species of Coccolithophores and found phagocytosis to be lower than 1% over 2 days. (Ref: Detection of Phagotrophy in the Marine Phytoplankton Group of the Coccolithophores (Calcihaptophycidae, Haptophyta) During Nutrient-replete and Phosphate-limited Growth - https://pubmed.ncbi.nlm.nih.gov/32233088/)

3. How does the life phases -haploid vs diploid affect phagotrophy and could the authors speculate as to why?

4. Does the night time represent a nutrient depleted condition, as photoautotrophy will not be possible? Despite nutrient-poor condition, the phagotrophy percentage remains low (~2 %) in the current study, where is the organism fulfilling most of its nutrient requirement?

5. Instead of beads, using organisms that can represent natural prey for the coccolithophores will be more relevant.

6. Mixotrophy is well known in coccolithophores, could the authors comment on the novelty of this study and how small percentages of phagotrophy mean a different nutrient strategy and how it affects the ocean ecosystem?

Comments on the Quality of English Language

English seems to be okay.

Author Response

Comments 1:

Emiliania huxleyi was named Gephyrocapsa huxleyi as mentioned in a recent publication (https://pubmed.ncbi.nlm.nih.gov/37983837/). Please check and rectify if needed to avoid confusion later.

Response 1:

It is really true as the reviewer suggested that the name of Emiliania huxleyi has been changed to Gephyrocapsa huxleyi. We have revised it in our paper.

Comments 2:

How does the current study agree or disagree with a previous study using fluorescently labelled bacteria to assess phagocytosis in different species of Coccolithophores and found phagocytosis to be lower than 1% over 2 days. (Ref: Detection of Phagotrophy in the Marine Phytoplankton Group of the Coccolithophores (Calcihaptophycidae, Haptophyta) During Nutrient-replete and Phosphate-limited Growth - https://pubmed.ncbi.nlm.nih.gov/32233088/)

Response 2:

The previous study held the same view with us that both the haploid and diploid strains showed the phagocytosis capacity. However, their results showed the low percentage (<1% of the population contained prey at all time points over 2 days). After careful examination, we think the reason may lie in the different method we used. The main difference is that the prey we use is fluorescent beads, which could not be digested and so provide a stable fluorescence signal. However, that makes it impossible to use it in the long-time experiment. For that reason, we set our incubation time to 2 h, while they use the FLB (fluorescently labeled bacteria) for 48 h incubation. There are some other differences in method. In their study, they use epifluorescence microscopy for the percentage calculation, while we treated the sample to reduce adhesion and avoid beads being stuck by coccoliths and use the flow cytometry to calculate, which examine more cells to make the percentage more accurate.

Comments 3:

How does the life phases -haploid vs diploid affect phagotrophy and could the authors speculate as to why?

Response 3:

We speculated that the mobility ability could be one of the reasons why they both showed phagotrophy differences. The haploid phase possesses two flagella, which means that it is more likely to approach the prey actively.

Comments 4:

Does the night time represent a nutrient depleted condition, as photoautotrophy will not be possible? Despite nutrient-poor condition, the phagotrophy percentage remains low (~2 %) in the current study, where is the organism fulfilling most of its nutrient requirement?

Response 4:

1. we considered it as a tough condition, not in nutrient but in the light regimes.

2. In our paper, we would like to the phagocytosis as one of the potential survival strategies rather than the growth strategy. Still, as we mentioned in the first part of discussion, long-term phagocytosis experiments still need to be done to find out if the phagocytosis is enough to support its survival and growth.

Comments 5:

Instead of beads, using organisms that can represent natural prey for the coccolithophores will be more relevant.

Response 5:

Thank you for the advice. In the future, we possibly use it to test the contribution of phagocytosis to the carbon fixation.

Comments 6:

Mixotrophy is well known in coccolithophores, could the authors comment on the novelty of this study and how small percentages of phagotrophy mean a different nutrient strategy and how it affects the ocean ecosystem?

 Response 6:

1. Compared to the other papers, we combine both flow cytometry and fluorescence microscopy to test the exact different phagocytosis percentages, which makes the conclusion more concrete.

2. Firstly, the phagocytosis percentage in our results doesn’t equal to the carbon source proportion, so further studies on the effect of phagocytosis to the energy needs to be done. Also, unlike the growth strategy, the survival strategy requires less energy, which could probably be supported by the phagocytosis.

3. Phagocytosis, as a potential carbon source, helps marine organisms utilize POC in the seawater, which accelerate the flow of energy and substance. However, instead of the ecological effect, we would like to attribute its contribution more to  the survival of this species.

Reviewer 2 Report

Comments and Suggestions for Authors

Title: Phagocytosis in Marine Coccolithophore Emiliania huxleyi: A Comparative Study of Calcified and Non-Calcified Strains

The article explores the phagocytic mechanisms between calcified diploid and non-calcified haploid strains of Emiliania huxleyi, and it also shed light on how these organisms acquire nutrients.
The present work has investigated the phagocytosis rates under different light conditions and ploidy levels, which provide useful insights in understanding their ecological implications. This paper mainly focuses on explaining how phagocytosis is involved in the uptake of nutrients by coccolithophores and the effect that calcification may have on their feeding habits. The analysis was successful due to application of robust statistical techniques as well as a clear experimental plan.

The paper clearly addresses an important aspect of coccolithophore biology- namely, phagocytosis in different strains of Emiliania huxleyi. Introducing the novel concept of comparing between calcified and non-calcified strains for better understanding their nutrient acquisition strategies helps contribute to knowledge about mixotrophy among marine phytoplankton. The study was designed appropriately to test the hypothesis with well described methods for reproducibility.

The most remarkable feature of the paper is its well organization and clarity in presentation of the research results. In any case, there are points where provision of more detailed reasoning or inclusion of extra control measures could enhance the study. For instance, specifying the particular phagocytosis assays used and possible limitations to the methodology would make these results more robust. Alternatively, a broader understanding of these findings might be derived from discussing how environmental factors influence phagocytosis rates.
This review topic is significant because it fills up a knowledge gap on phagocytosis capabilities among different strains of Emiliania huxleyi. The paper also effectively guides us through differences between calcified and non-calcified strains with regard to nutrient acquisition strategies which adds to our understanding about mixotrophy in marine phytoplankton. Most references cited are recent and relevant to the subject matter hence forming a good base for this study. However, further elaboration on ecological significance of these findings and their relationship with wider marine ecosystem dynamics may increase its impacts.

In general, the paper presents considerable findings regarding phagocytosis efficiency and physiological variations among E. huxleyi strains that aid in the understanding of coccolithophore ecology. Incorporating long-term experiments, energy estimation and elaborate controls into the methodology as well as broadening discussion about ecological ramifications of these physiological disparities would make the study more impactful. Furthermore, this research could be strengthened by acknowledging its drawbacks and elaborating further on experimental controls to strengthen the scientific balance of the paper.

Specific Comments:
Line 203: The statistical analysis methods employed in the research should be described in detail including the specific variables used and why t-tests and ANOVA were chosen. Giving assumptions made, and significance levels got would increase transparency of data analysis.

Figure 1: Figure 1 can have some more labels or annotations that show conditions under which the phagocytosis assays were conducted. The duration of experiments, light intensities and any controls included should be provided so as to get an accurate interpretation of results.

Table 1: Although Table 1 shows phagocytosis percentages for different strains in both dark and light conditions, additional data on abiotic factors like temperature, salinity, and nutrient concentrations during the experiment can help us appreciate how these impact on rates of phagocytosis observed.

Line 259: This is a very interesting discussion on long-term implications of the phagocytosis experiments. Nonetheless, it would be good if more could be said about how various light intensities affect rate of phagocytosis as well as how phagocytosis relates with photosynthesis in different Emiliania huxleyi strainsLine

Figure 4: The legend in Figure 4 provides helpful information about the probability of achieving the given levels for the percentages of phagocytosis. The results would have more value if the sample sizes, the number of replicates, and the statistical approach used to compare the calcified and control strains were further annotated.

Line 309: Explaining phagocytosis as ‘evening out nutrients that are particulate in form’ by predatory or mixotrophic organisms is fine, but some additional detail on the specific genes and proteins involved in the process of phagocytosis in E huxleyi, and how they compare between calcified and non-calcified strains, would give us a better sense of how nutrient acquisition strategies are mediated at a molecular level.

If you reply to all of these comments, you reinforce the article and make it more scientifically rigorous and reader-friendly – improving the contribution that microbiology will make to marine biology.

Author Response

Comments 1:

Line 203: The statistical analysis methods employed in the research should be described in detail including the specific variables used and why t-tests and ANOVA were chosen. Giving assumptions made, and significance levels got would increase transparency of data analysis.

Response 1:

We are very sorry for the unclear expression in the statical analysis. We have rewritten this part according to the Reviewer’s suggestion (Line 202-213).

Comments 2:

Figure 1: Figure 1 can have some more labels or annotations that show conditions under which the phagocytosis assays were conducted. The duration of experiments, light intensities and any controls included should be provided so as to get an accurate interpretation of results.

Response 2:

It is really true as Reviewer suggested that more details should be added. However, Figure 1 showed the features of two strains in the stocking conditions, not in the phagocytosis experiment condition. All the details are written in the method part. In case what you mentioned is Figure 4, we add more details in that figure. At all events, we add more details in Figure S1, S2 and appreciate your suggestion. (Line 223-227, Line 263-267, supplementary Figure S1 and S2)

Comments 3:

Table 1: Although Table 1 shows phagocytosis percentages for different strains in both dark and light conditions, additional data on abiotic factors like temperature, salinity, and nutrient concentrations during the experiment can help us appreciate how these impact on rates of phagocytosis observed.

Response 3:

We appreciate your advice. However, this paper doesn’t include any table. I wonder if you mean Figure S1 and S2 or suggest us to put the exact percentages in one table. We have revised Figure S1 and S2.

Comments 4:

Line 259: This is a very interesting discussion on long-term implications of the phagocytosis experiments. Nonetheless, it would be good if more could be said about how various light intensities affect rate of phagocytosis as well as how phagocytosis relates with photosynthesis in different Emiliania huxleyi strainsLine

Response 4:

Thank you again for your positive comments and valuable suggestions. We have added more information in this part. However, the phagocytosis was less studied in this species and we didn’t find similar researches as you suggested. As a result, we discussed the relationship between photosynthesis and phagocytosis among more phytoplankton. (Line 277-283)

Comments 5:

Figure 4: The legend in Figure 4 provides helpful information about the probability of achieving the given levels for the percentages of phagocytosis. The results would have more value if the sample sizes, the number of replicates, and the statistical approach used to compare the calcified and control strains were further annotated.

Response 5:

We appreciate your suggestions and have rewritten this part according to the suggestions. (Line 263-267)

Comments 6:

Line 309: Explaining phagocytosis as ‘evening out nutrients that are particulate in form’ by predatory or mixotrophic organisms is fine, but some additional detail on the specific genes and proteins involved in the process of phagocytosis in E huxleyi, and how they compare between calcified and non-calcified strains, would give us a better sense of how nutrient acquisition strategies are mediated at a molecular level.

Response 6:

Thank you for your advice. Explaining the phagocytosis at a molecular level is helpful to perfect our paper. We have already added more details as suggested by the reviewer. (Line 324-328)

Reviewer 3 Report

Comments and Suggestions for Authors

In this study, the authors performed a series of experiments on the phagocytosis of calcified diploid and non-calcified haploid strains of the coccolithophore Emiliania huxleyi in light and darkness. As a result, when comparing the phagocytosis of these strains, a higher level was revealed in the case of the non-calcified haploid strain in the dark. Thus, various feeding strategies of the coccolithophore at different stages of life and the stage of calcification are revealed. The work is of fundamental importance. This study sheds light on the potentially different strategies that coccolithophores have at different life stages in twilight zone conditions and during adverse climate change events, which also determines its relevance. The work was carried out at a good methodological level; an interesting methodological approach was used to measure the level of phagocytosis; characteristic fluorescent signals from ingested beads were determined using fluorescence microscopy and flow cytometry. The study will contribute to the understanding of the marine carbon cycle.

The article as a whole is written in clear language and well organized. however, there are small inaccuracies and errors in the text, so I wish the authors to edit the text carefully.

Literature references should be numbered and given in square brackets as they are mentioned. Although there are a small number of links to sources published in 2020-22, why are there no links to works from 2023? Is it possible to add a few very recent sources in the introduction?

L 198-199 … The phagocytosis percentages (PP) were calculated.. This is not a percentage, but a share. To get percentages you need to multiply by 100.

L 203-206. Have the authors verified that the data is normally distributed and that parametric statistics and the listed tests can be applied?

L 280… The results revealed…

L 333 ….the role of E. huxleyi as a consumer

L 335      at the community level

L 338… under different light regimes, thus..

L 347 … the cells sink 200 to

L 356 … possessed a higher

L 357 … diploid calcified phase

L 358 … transform into the haploid non-calcified phase of life cycle..

L 364 … for understanding

L 377 … osmotrophic pathway

Figure 5 caption…. The particle organic carbon can be phagocytosed to a higher percentage in the non-calcified phase as compared to the calcified phase in the dark as compared to the light condition in our study.  

Rephraze as “In our investigation, the particle organic carbon can be phagocytosed to a higher percentage in the non-calcified phase than the calcified phase in the dark as compared to the light condition.

After making corrections and checking the English language, the work can be accepted for publication.

Comments on the Quality of English Language

The article as a whole is written in clear language and well organized. however, there are small inaccuracies and errors in the text, so I wish the authors to edit the text carefully.

Also, clarification on statistics is required, adding to the introduction. I listed minor comments for the authors. After making corrections and checking the English language, the work can be accepted for publication.

Author Response

Comments 1:

Literature references should be numbered and given in square brackets as they are mentioned. Although there are a small number of links to sources published in 2020-22, why are there no links to works from 2023? Is it possible to add a few very recent sources in the introduction?

Response 1:

We sincerely appreciate the valuable comments. We have checked the literature carefully, use the numbered citation form (Line 605-768) and added more references into the INTRODUCTION part in the revised manuscript( Line 59-60, 72-74).

Comments 2:

L 198-199 … The phagocytosis percentages (PP) were calculated.. This is not a percentage, but a share. To get percentages you need to multiply by 100.

Response 2:

Thank you for the suggestions. We have revised the formula. (Line 198)

Comments 3:

L 203-206. Have the authors verified that the data is normally distributed and that parametric statistics and the listed tests can be applied?

Response 3:

Thank you for the suggestions. We performed the Shapro-Wilk test on the cell diameter, which proved its normal distribution. As for the phagocytosis percentages, for the resource limitation, the general parameters are taken three parallel samples, so it is statistically significant. (Line 202-213)

Comments 4:

L 280… The results revealed…

Response 4:

Thank you for the suggestions. We have revised this sentence. (Line 292)

Comments 5:

L 333 ….the role of E. huxleyi as a consumer

Response 5:

Thank you for the suggestions. We have revised this sentence. (Line 344)

Comments 6:

L 335      at the community level

Response 6:

Thank you for the suggestions. We have revised this sentence. (Line 346)

Comments 7:

L 338… under different light regimes, thus..

Response 7:

Thank you for the suggestions. We have revised this sentence. (Line 348)

Comments 8:

L 347 … the cells sink 200 to…

Response 8:

Thank you for the suggestions. We have revised this sentence. (Line 357)

Comments 9:

L 356 … possessed a higher

 Response 9:

Thank you for the suggestions. We have revised this sentence. (Line 365)

Comments 10:

L 357 … diploid calcified phase

Response 10:

Thank you for the suggestions. We have revised this sentence. (Line 366)

Comments 11:

L 358 … transform into the haploid non-calcified phase of life cycle..

Response 11:

Thank you for the suggestions. We have revised this sentence. (Line 367)

Comments 12:

L 364 … for understanding

Response 12:

Thank you for the suggestions. We have revised this sentence. (Line 372)

Comments 13:

L 377 … osmotrophic pathway

Response 13:

Thank you for the suggestions. We have revised this sentence. (Line 385)

Comments 14:

Figure 5 caption…. The particle organic carbon can be phagocytosed to a higher percentage in the non-calcified phase as compared to the calcified phase in the dark as compared to the light condition in our study. 

Rephraze as “In our investigation, the particle organic carbon can be phagocytosed to a higher percentage in the non-calcified phase than the calcified phase in the dark as compared to the light condition.”

 Response 14:

Thank you for the suggestions. We have revised this part. (Line 393-397)